# Topology Optimization of a Femoral Stem in Titanium and Carbon to Reduce Stress Shielding with the FEM Method

**Mario Ceddia** [1,*], **Bartolomeo Trentadue** [1] , **Giuseppe De Giosa** [2] **and Giuseppe Solarino** [3]

1 Department of Mechanics, Mathematics and Management, Politecnico di Bari University, 70125 Bari, Italy; bartolomeo.trentadue@poliba.it
2 Department of Translational Biomedicine and Neuroscience, University of Bari, 70125 Bari, Italy; degiosa_giuseppe@libero.it
3 Orthopaedic Unit, Department of Neuroscience and Organs of Sense, Faculty of Medicine and Surgery, University of Bari "Aldo Moro", Policlinico-Piazza G. Cesare, 11, 70124 Bari, Italy; giuseppe.solarino@uniba.it
* Correspondence: marioceddia1998@gmail.com

**Abstract:** Arthroplasty is commonly performed to treat advanced osteoarthritis or other degenerative joint conditions; however, it can also be considered for young patients with severe joint damage that significantly limits their functionality and quality of life. Young patients are still at risk of aseptic mobilization and bone resorption due to the phenomenon of stress shielding that causes an uneven distribution of tensions along the femoral contact surface prosthesis. This phenomenon can be limited by choosing the material of the prosthesis appropriately or by varying its stiffness, making sure that its mechanical behavior simulates that of the femur as much as possible. The aim of this study is to evaluate the mechanical strength of a prosthesis optimized both in shape and material and compare the results with a standard titanium prosthesis. **Methods**: Through three-dimensional modeling and the use of finite element method (FEM) software such as ANSYS, the mechanical behavior of traditional prosthesis and prosthesis optimized topologically respecting the ASTM F2996-13 standard. **Results**: With topological optimization, there is a stress reduction from 987 MPa to 810 MPa with a mass reduction of 30%. When carbon fiber is used, it is possible to further reduce stress to 509 MPa. **Conclusions**: The reduction in stress on the femoral stem allows an optimal distribution of the load on the cortical bone, thus decreasing the problem of stress shielding.

**Keywords:** hip prosthesis; finite element analysis; composite material; stress shielding; topological optimization

## 1. Introduction

### 1.1. State of the Art of Arthroplasty

Patients who undergo arthroplasty live longer, more active lives and receive joint replacements at an increasingly early age [1,2]. As a result, total hip arthroplasty (THA) in this younger population is subject to increased physiological stress, and the continued existence of prosthetics is a persistent clinical concern. For older patients, implant survival rates for THA exceed 95% after 10 years but subsequently drop to 80–85% after 18 years [3,4]. This is in contrast with younger patients undergoing THA, where reported survival rates drop to 72–88% at 10 years for patients under 60 and 68% for patients under 55. In the past, stem breakage, coarse acetabular wear, or total fatigue failure of metal femoral stems were considered the main causes of THA failure [2].

### 1.2. Current Situation of Arthroplasty

Nowadays, aseptic loosening, related to osteolysis of wear debris created at the joint and micro-movements at the bone–implant interface, is considered the predominant cause of revision for metal hip replacements. When revision surgeries are performed, bone resorption, related to the difference in stiffness between the implant and the host tissue (i.e.,

the femur), is a concern as the host bone becomes weaker and less able to receive larger hip replacements. This phenomenon is known as "Stress Shielding" and is often seen with metal implants. In addition, questor stress shielding could contribute to the concentration of stress at the bone–implant interface leading to micromovements, as recently indicated by a simulation study of the fine elements of the stems of total hip replacement (THP) [3]. Due to the difference between the rigidity of the prosthesis material (such as alumina ceramics, titanium alloy, zirconium niobium alloy, and carbon fiber composite materials) and bone stiffness, stress shielding will lead to the loosening of the stem–femur connection and even fractures and other serious consequences [2–4]. The formation of a porous structure provides a better solution to the problem of this effect [5]. A further solution to reduce bone loss associated with stress shielding is the use of implants with an overall reduction in flexion stiffness. This can be achieved in two ways, by using composite materials (CF) with rigidity and mechanical characteristics closer to that of the human femur [6,7] or by using topological optimization, which aims to modify the stiffness of the stem based on the von Mises results.

### 1.3. Stem Optimization to Reduce Stress Shielding

This research better helps the understanding of the mechanisms of failure and allows the designer to be able to create an optimized prosthesis in relation to clinical parameters. In addition to the well-known structural and mechanical properties [8,9], carbon fibers have some biocompatible properties that have been recognized clinically [10,11] via animal research and experiments in the laboratory [12–17]. Carbon fiber is lightweight with a density of $1.6/2.2 \text{ g/cm}^3$ [8,9,17] compared to the density of compact bone at $2.0 \text{ g/cm}^3$ [18,19]. Small-diameter, high-strength, high-modulus carbon fibers can be shaped to fit into complex curved spaces for multiple variations in applied use [9–12]. Due to the potential benefits of developing high-strength biomaterials with a density closer to the bone for better stress transfer and electrical properties that improve tissue formation, a test model of rat animal tibia implantation in vivo was used to demonstrate possible biocompatible improvements for carbon fiber in reinforced polymer matrix composite material [20]. Results showed that the carbon-fiber-reinforced composite stimulated osseointegration within the bone marrow of the tibia, which measured as a percentage of the bone area (PBA) to a large extent compared to titanium alloy. Adam et al. [21] conducted a human clinical study on a smooth-surfaced press-fit carbon fiber hip replacement and observed that the modulus of a carbon stem is about three times lower than metal stems and closer to the cortical bone modulus (ranging from 12 GPa to 20 GPa). This study revealed that the carbon fiber composite material has the mechanical properties to withstand the physiological stress of a hip joint but that insufficient bone fixation due to the smooth surface of the prosthesis caused early loosening of the implant. Therefore, the tissue response of a carbon stem with hydroxyapatite coating [22] was evaluated by observing a high degree of bone apposition to carbon fiber composite implants coated with HA. The lack of short-term inflammation and adverse tissue response to carbon fiber composite implants that support the ongoing evaluation of this composite technology for use in THA were also studied. Moreover, as reported by Campbell et al. [6], mechanical fatigue failures occurred after about 10 ^4 cycles for the maximum fatigue load of 22 kN (i.e., 123 MPa), 10 ^5 cycles for the maximum fatigue load of 20 kN (i.e., 112 MPa), 10 ^6 cycles for the maximum fatigue load of 18 kN (i.e., 101 MPa), and close to 10 ^7 cycles and more for the maximum fatigue load of 17 kN (i.e., 95 MPa). These results show fatigue life at least six times higher than the load levels recommended by the ASTM standards [22]. Aseptic loosening of the implant is mainly influenced by the phenomena of bone resorption that are revealed in certain regions of the femur when a prosthesis is introduced. As a result, bone resorption appears due to stress shielding, i.e., the decrease in the level of stress in the implanted femur caused by the significant carrying load of the prosthesis due to its increased rigidity. A strategy based on the topological optimization (TO) of maximum stiffness is used for nonlinear finite element static (FE) analyses of femur–implant assembly, with the aim

of reducing stress shielding in the femur and providing guidelines for hip replacement redesign. This is achieved by employing extreme accuracy for both three-dimensional reconstruction of femur geometry and material property maps assigned as explicit local density functions. The topological optimization design method is based on a methodology that allows for determining the optimal distribution of material within a structure to maximize performance and minimize weight. In the field of femoral prostheses, topological optimization can be used to design an optimal shape of the prosthesis, ensuring maximum strength and rigidity with the least possible use of material. It allows to form a continuous mechanical transmission path, which can better solve the problem of reducing the shielding of stresses. In this work, both methods will be compared, and the results will be weighed against a titanium prosthesis as a result of the finite element method adopted in this study. FEM (Finite Element Method) analysis is an engineering analysis methodology used to evaluate the structural behavior and performance of a femoral prosthesis. This computational method is based on the discretization of geometry into finite elements, which are mathematically modeled to represent the mechanical behavior of the material and the response to stresses. The hypothesis behind this study is that the decrease in stem stiffness, through the modification of the section or material, leads to a reduction in stress shielding. Thanks to this study, topological optimization is related for the first time to the optimization obtained by choosing a material such as carbon fiber.

## 2. Materials and Methods

The prostheses were modeled using Autodesk Inventor, as seen in Figure 1. Autodesk Inventor is a 3D modeling and mechanical design software widely used for creating part and assembly models. The model was then transferred to the ANSYS Workbench software to analyze the system. The properties of the prosthesis material (Ti6Al4V) were assumed to be linear homogeneous and isotropic. Table 1

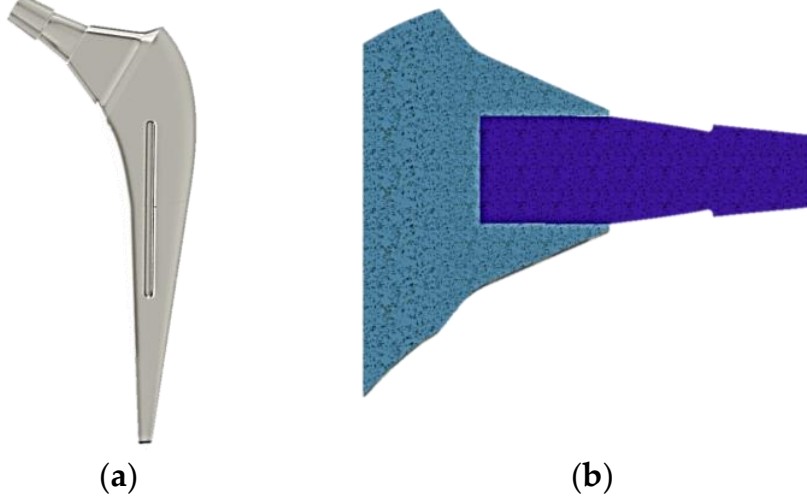

(a)          (b)

**Figure 1.** (**a**) 3D model of the prosthesis; (**b**) section showing the insertion of the head by means of conical connection.

**Table 1.** Mechanical properties of the material for the femoral stem [23,24].

| Material | Density (Kg/m$^3$) | Young Modulus (GPa) | Poisson's Ratio | Ultimate Strength (MPa) | Yeld Strength (MPa) |
|---|---|---|---|---|---|
| Ti6Al4V | 4500 | 110 | 0.32 | 900 | 800 |

*Meshing*

The ANSYS was used to generate the mesh with the number of nodes and elements for the present study at 629.995 and 449.179, respectively. The mesh of the implant model used ten-knot tetrahedral elements (Figure 2), and the tetrahedral elements were carefully inserted into the irregular form of three-dimensional geometry. The optimal selection of the mesh is an important parameter to obtain accurate results. Moreover, a mesh-independent study was carried out to select the appropriate mesh size. Figure 3 demonstrates the mesh independence test, which shows the relationship between the n of the elements and von Mises stress. The element size of 0.71 mm is optimal for the present study since, as seen in Figure 3, this dimension allows to have a stable von Mises stress at about 700 MPa.

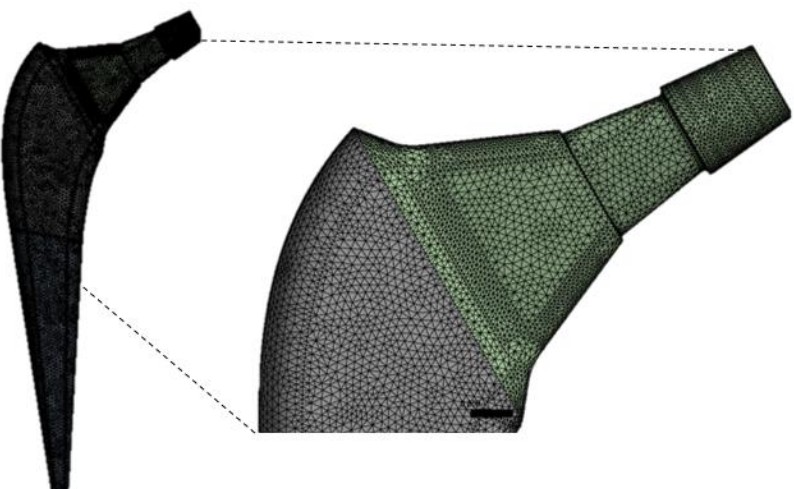

**Figure 2.** Meshed model of implant.

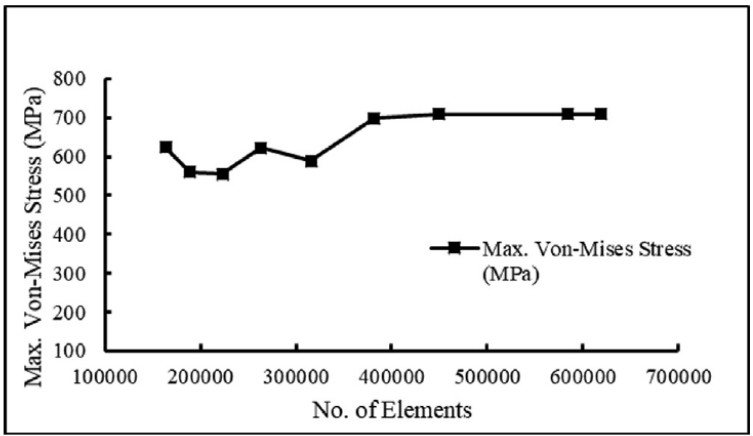

**Figure 3.** Mesh independency test [22].

The boundary conditions involve the application of constraints and forces. This analysis is performed using boundary conditions according to ASTM F2996-13 [22]. The constraint conditions in the design of a femoral prosthesis refer to the restrictions or constraints imposed on the prosthesis to ensure its stability and integration with the surrounding bone. These constraint conditions are determined by the type of implant and its anatomical location in the human body [24]. The dynamic load of the average walk is applied to the femoral stem from which they were taken [25]. In further analysis, the load deriving from a monopodalic support is also considered [26]. Any type of torsional, rotational, or tensile force is not considered, and the distal end areas are kept fixed (Figure 4). For the analysis of homogeneous models with multiple material variations, the ANSYS FEA code is used as a tool to solve the finite element model.

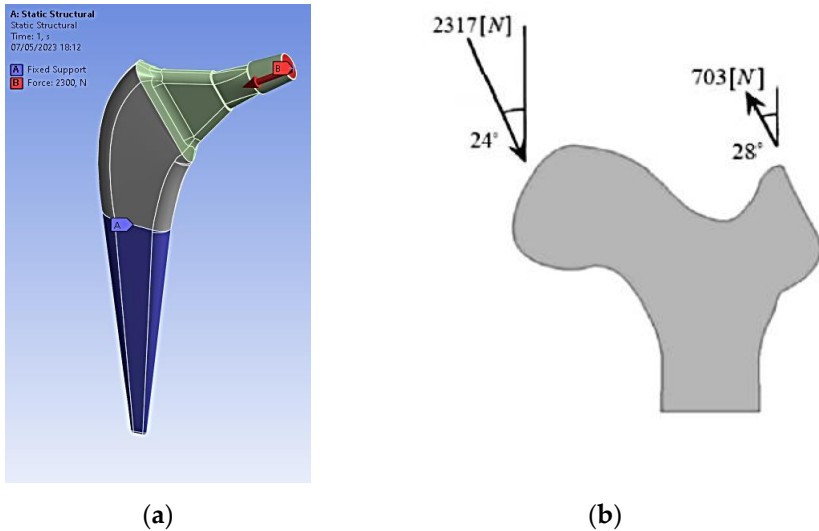

(**a**)　　　　　　　　　　　(**b**)

**Figure 4.** (**a**) Constraint and loading condition according to ASTM F2996-13; (**b**) monopodal load application [26].

## 3. Results

The distribution of tensions is one of the important information that can be obtained through the finite element analysis (FEM) of a femoral prosthesis. Von Mises stress is one of the most common parameters used to assess the state of stress within a structure.

Von Mises stress is a measure of equivalent stress that takes into account the main stresses in a particular region or component. This parameter considers the stresses in all directions and provides a unique value that represents the maximum equivalent stress in the analyzed region. The von Mises results for a femoral prosthesis can be represented through a color map or a graph showing the distribution of tension in the entire prosthesis or in specific regions of interest. Different colors can be used to indicate different levels of tension, allowing a visual assessment of high and low-stress areas. The von Mises results are important because they help identify critical areas subject to high stress, which could be subject to potential failure or deformation. This information can be used to evaluate and optimize the design of the prosthesis, identifying areas that require further modification or reinforcement. For the applied load according to ASTM F2996-13, and that applied considering the mono-breech load, Figure 5 shows the von Mises distribution on the titanium stem.

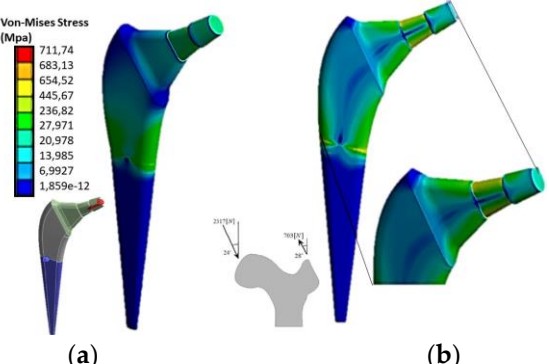

(**a**)　　　　　　　　　　　(**b**)

**Figure 5.** Von Mises stress distribution (**a**) load according to ASTM F2996-13; (**b**) Mono-breech load.

As can be seen in the first case, there are about 700 MPa of stress on the upper area of the stem, compared to the second case in which about 987 MPa of stress are reached, a value higher than the breaking load of the material. Therefore, the mono-breech load condition is the most critical to which a femoral prosthesis can be subjected.

### 3.1. Topology Optimization

The goal was to reduce weight by at least 30%; to do this, the ANSYS software, based on the assigned constraint and load conditions and shape retention conditions on the implant neck and in the area of contact with the femur, performed 87 iterations (Figure 6) and, eventually, based on the maximum von Mises stresses, created a topologically more efficient form (Figure 7).

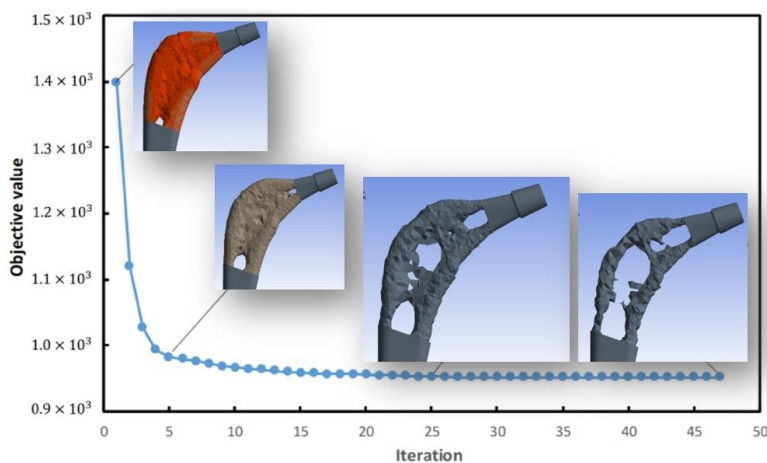

**Figure 6.** Topological configuration, solid model of the femoral prosthesis with its iteration cycle to obtain it.

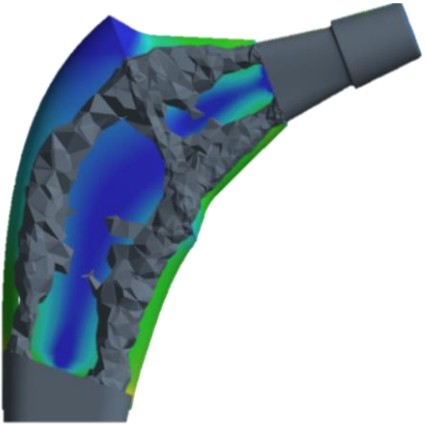

**Figure 7.** Topological optimization of the stem based on von Mises stress.

Once the ANSYS finite element software resulted in the most topologically efficient form, the new geometry was simulated by performing a new mesh, leaving the mesh settings used for the previous prosthesis unchanged. It is subsequently, under the same condition of constraint and monopodialic load studied previously; we went to study the von Mises stress distribution, comparing it with the full prosthesis (Figure 8).

The results shown in Figure 9 demonstrate how the topological optimization carried out on the same material (Titanium) led to a reduction of 30% in mass, and the maximum stress went from 987 MPa to about 810 Mpa, a value more than acceptable for the material.

The maximum tension, as is observed in Table 2, is concentrated in the areas that have a reduced section. This is particularly seen on the neck of the stem, where stress values between 550 MPa and 660 MPa are reached. Because the yield stress is 800 Mpa, it can be assumed that there are no critical issues.

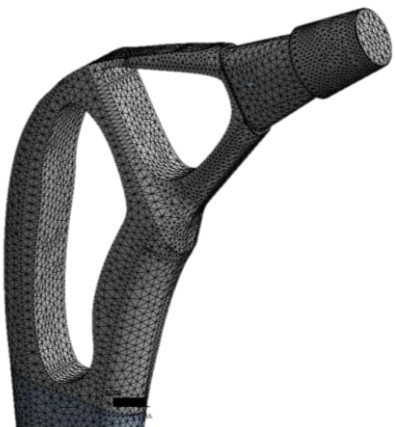

**Figure 8.** Optimized prosthesis mesh.

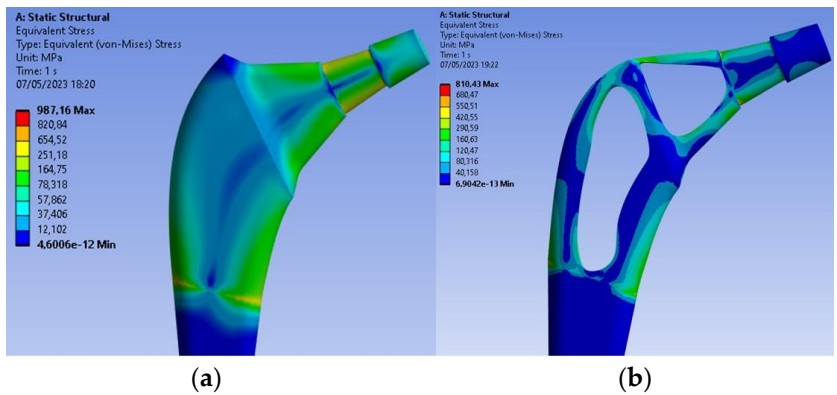

|                                 |   (**a**)   |   |   (**b**)   |

**Figure 9.** Von Mises stress distribution (**a**) solid stem; (**b**) Topologically optimized hollow stem.

**Table 2.** Summary of the main results obtained from the first study.

|  | Load: ASTM F2996 | Load: Monopodalic |  |
|---|---|---|---|
| Von Mises Stress (Mpa) | 700 | **987** | Full stem (Ti6Al4V) |
|  |  | **810.43** Mass −30% | Topologically optimized hollow shaft (Ti6Al4V) |

### 3.2. Reduction in Stress Shielding with Carbon Fiber Prostheses

As anticipated in the introduction, a potential solution to reduce bone loss associated with stress shielding is the use of implants with an overall reduction in flexion stiffness. In this regard, a femoral component manufactured from a CF composite with an intermediate polymer layer and a hydroxyapatite (HA) coating for cementless fixation was developed (Figure 10).

Hydroxyapatite is the most documented calcium phosphate and can be used in a bulk form as a coating and/or cement [27,28]. This material can be classified according to its porosity, phase, and processing method. It has excellent biocompatibility and is able to promote osteoconduction and osseointegration. As a result of its excellent conductive and favorable bioactive properties, it is widely preferred as a biomaterial of choice in both dentistry and orthopedics [29,30]. While one of the advantages of carbon fiber is that it is biocompatible [31–34], promotes bone growth on the surface, and has mechanical characteristics similar to cortical bone, there is a reduction in stress shielding. Under the same constraint and load conditions shown in Figure 4a, the results on the von Mises stress distribution for the rod topologically optimized in composite material are reported in Figure 11.

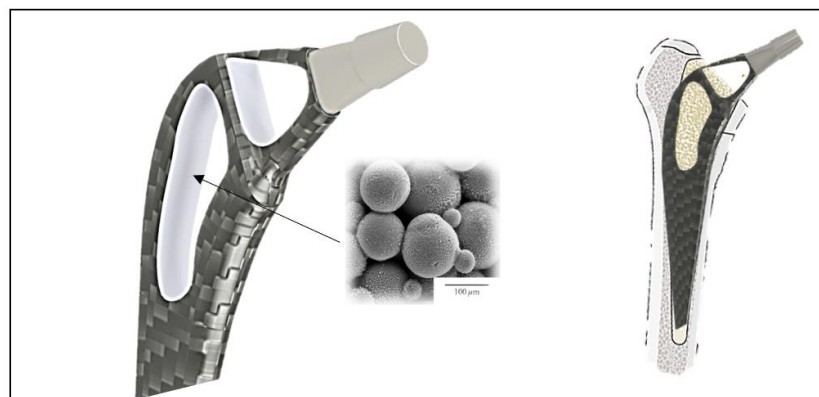

**Figure 10.** Topically optimized carbon fiber stem with inner lining in (HA).

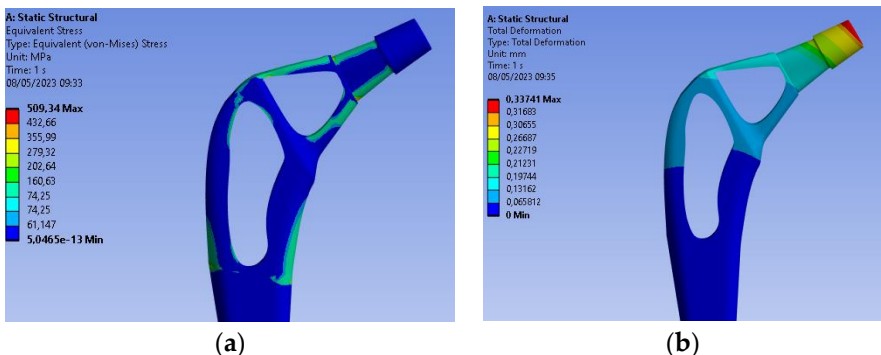

**Figure 11.** (**a**) Von Mises stress; (**b**) Total deformation on the stem.

From the results Figure 12, it is seen how the use of carbon fiber allows for decreasing the von Mises stress for the high mechanical strength of the fiber; in fact, it goes from 810 MPa (Titanium) to 509 MPa (Carbon); also, the total deformation decreases from 0.51 mm (Titanium) to 0.33 mm (Carbon). Therefore, this decrease in both stress and deformation leads to an increase in stiffness of the entire stem with advantages on stress shielding.

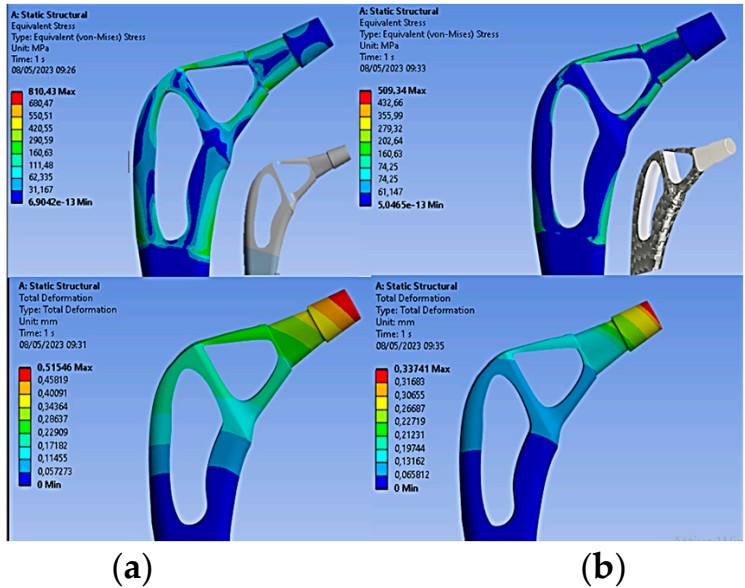

**Figure 12.** Results of von Mises stress (**a**) and total deformation (**b**) for the topologically optimized stem in Titanium, on the left, and in composite material, on the right.

## 4. Discussion

The most important finding of this study is that topological optimization can be used to design femoral and selectively hollowed-out stems with higher compliance for less stress shielding. In the field of femoral prostheses, topological optimization can be used to design and optimize the shape and structure of the prosthesis itself. The goal of topological optimization in femoral prostheses is to reduce the weight of the prosthesis while maintaining its necessary strength and rigidity [35]. This can result in lighter, more comfortable, and more functional prostheses for patients. The topology optimization process typically involves several steps:

- Definition of requirements: the desired performance requirements for the femoral prosthesis are defined as strength, stiffness, and maximum permissible weight;
- Creating a CAD Model: a computer-aided design (CAD) model of the prosthesis is created, representing its geometry and structure;
- Enforcement of restrictions: the necessary restrictions and constraints are applied, such as physical space limits, location of holes for fixing screws, or other specific considerations;
- Definition of the design region: the area in which the distribution of the material for the prosthesis can be varied is defined;
- Application of optimization: Through topology optimization algorithms, different configurations are explored to determine the optimal distribution of the material. The goal is to reduce weight while maintaining the required performance;
- Analysis and verification: optimized configurations are analyzed through simulations or structural analysis to evaluate performance and feasibility;
- Prosthesis realization: once an optimized design is achieved, the prosthesis can be made using additive manufacturing technologies or other manufacturing methods;
- Topological optimization can help improve the performance and efficiency of femoral prostheses, but it is important to emphasize that the process requires specialized skills and resources.

The use of carbon fiber composite material has allowed for obtaining further optimization since the composite having mechanical characteristics in terms of rigidity comparable to those of the bone allowed for further reducing the shielding of the stresses. Moreover, as shown in [36], the carbon composite is biocompatible, and the bone affixing on this material is 15% greater than the titanium alloy, and the use of a hydroxyapatite coating allows this percentage to be increased to 37% [37]. Carbon fiber femoral prostheses offer several advantages over traditional metal prostheses. First, carbon fiber is lightweight, which means the prosthesis will weigh less than a metal prosthesis. This can help reduce the load on the rest of the limb and improve patient mobility. In fact, thanks to their combination of lightness and strength, carbon fiber femoral prostheses can reduce stress and pressure exerted on the surrounding bone. This can help prevent bone deterioration, reduce the risk of stress fractures, and improve bone integration with the denture. Lower risk of corrosion: Unlike metal alloy dentures, carbon fiber femoral dentures are not subject to corrosion. This means that they will not corrode or damage over time due to corrosive processes. This can contribute to the longer life of the prosthesis and reduce the need for revision surgery. Possibility of customization: Carbon fiber is a highly moldable and adaptable material, which allows the realization of highly personalized femoral prostheses. Carbon fiber prostheses can be designed to adapt to the specific anatomical and functional needs of the patient, ensuring better adaptation and better performance. In addition, carbon fiber is a rigid and durable material, which allows for making thinner and less invasive dentures. This means that the prosthesis can be implanted with less surgical invasiveness and that the patient can benefit from faster rehabilitation. Carbon fiber femoral prostheses are designed to anatomically adapt to the shape of the femur, providing greater stability and better coupling with the surrounding bone. This can help reduce the risk of complications such as unscrewing or sagging the denture. However, it is important to note that the use of carbon fiber femoral implants may not be indicated for all patients. The choice of prosthesis depends on several factors, such as the general state of health of the patient, age, level

of physical activity, and the specific condition of the femur to be replaced. The coating adopted for the hydroxyapatite carbon prosthesis is widely used as a coating in femoral prostheses to promote osseointegration, i.e., the process of fusion of the prosthesis with the surrounding bone [38,39].

The hydroxyapatite coating on the surface of femoral prostheses offers several advantages:

- Better osseointegration: Hydroxyapatite creates a biocompatible interface between the bone and the prosthesis, facilitating the adhesion and growth of bone cells. This promotes the formation of a solid bond between the prosthesis and the surrounding bone, reducing the risk of developing complications such as sagging or relaxation of the prosthesis;
- Stimulation of bone regeneration: Hydroxyapatite can help stimulate bone regeneration due to its structural similarity to natural bone tissue. It promotes the deposition of new bone tissue around the prosthesis, increasing the stability and durability of the implant;
- Reduction in inflammation and infection: The hydroxyapatite coating can help reduce inflammation and the risk of infection. Its biocompatible surface reduces the unwanted immune response and can hinder bacterial adhesion to the surface of the prosthesis.

Improvement of mechanical properties: Hydroxyapatite can also improve the mechanical properties of the femoral prosthesis. By acting as a durable coating layer, it can help reduce wear and abrasion of the prosthesis, prolonging its durability.

Importantly, hydroxyapatite coating can be applied in different forms, such as porous coatings, spray coatings, or electrochemical coatings. The choice of method depends on the specifics of the prosthesis and the preferences of the orthopedic surgeon.

## 5. Conclusions

In conclusion, a topological optimization process was developed and applied to the design of the femoral stem, which included a modification of the geometry as a function of the von Mises stress distribution and a variation on the material, opting for a composite material that, as already shown in various studies, offers histological characteristics that make it implantable. This optimization has made it possible to solve a problem that leads to the failure of the implants or stress shielding. This is a phenomenon that occurs when the component inserted in the bone has a different stiffness, and therefore, the distribution of stresses is not dispersed evenly to the bone–implant interface, but tension concentrations lead to inflammatory phenomena on the bone, reducing its ability to create optimal osseointegration. The main results obtained in this study are reported in terms of von Mises tension.

**Von Mises stress**
Femoral stem in Titanium alloy: 987 Mpa;
Optimized Titanium alloy femoral stem: 810 Mpa;
Femoral stem in composite material: 509 Mpa.

Finite element analysis (FEM) used in this study is a numerical technique that is often used to evaluate the structural behavior of femoral prostheses.

Although FEA is a powerful analytical tool, it is important to remember that FEA results must be validated experimentally to ensure the reliability of the results. The lack of valid experimental validation may limit confidence in the results obtained through the FEA.

**Author Contributions:** Conceptualization, M.C. and B.T.; methodology, M.C.; software, M.C.; validation, B.T. and G.S. formal analysis, M.C. and B.T.; investigation, M.C.; resources, M.C. and B.T.; data curation, M.C.; writing—original draft preparation, M.C. and B.T.; writing—review and editing, M.C., B.T., G.S. and G.D.G.; visualization, B.T. and G.S.; supervision, G.S. and B.T.; project administration, G.S. and B.T. All authors have read and agreed to the published version of the manuscript.

**Funding:** This research received no external funding.

**Institutional Review Board Statement:** Not applicable.

**Informed Consent Statement:** Not applicable.

**Data Availability Statement:** All experimental data to support the findings of this study are available by contacting the corresponding author upon request.

**Conflicts of Interest:** The authors declare no conflict of interest.

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
