# Peer review of "Topology Optimization of a Femoral Stem in Titanium and Carbon to Reduce Stress Shielding with the FEM Method"

_jcs, doi:10.3390/jcs7070298_

Round 1

Reviewer 1 Report

The study was carried out by assessing the mechanical behaviour of prostheses optimised in terms of shape and material and comparing the results with standard titanium prostheses. The conclusions of the study have important implications for the design of joint prostheses. There are still some issues in the article that need to be revised, as follows:

1.The preamble is suggested to be divided into three paragraphs, the first part of the research background and significance, the second paragraph of the current situation of domestic and international research and the existing problems, and the third paragraph of the main research content of this paper.

2.The article has grammatical and spelling problems and lacks the necessary punctuation in some places.

3.Images can have their backgrounds removed to increase the viewability of the article.

4.In Table 1, Mpa should be changed to MPa and Gpa to GPa. It is suggested that the authors recheck the grammatical spelling throughout.

5.The conclusion section is too long and the author is advised to streamline it.

6.Suggested author reference Chang L, Wang H, Guo Y, et al. Experimental and numerical analysis of biomechanical effects in cervical spine positioning rotation manipulation[J]. International Journal for Numerical Methods in Biomedical Engineering, 2022: e3651. revised and with references from the last five years.

Moderate editing of English language required

Author Response

I have sent responses to you review

Reviewer 2 Report

The submitted article with the Manuscript ID: “jcs-2488585” and the title: “Topology optimization of a femoral stem in Titanium and Carbon to reduce Stress Shielding with the FEM method” presents an interesting topological optimization process applied to the design of the femoral stem. It extends the Von-Mises stress distribution function and includes a variation on the material opting. The developed computational method is based on discretizing geometry into finite element method (FEM) analyses using the software ANSYS to simulate the mechanical behavior of traditional prosthesis and prosthesis optimized topologically respecting the ASTM F2996-13 standard. The paper falls within the scope of the Journal. The presented study is certainly of interest to the readers of the Journal, the manuscript is well-structured, and the findings obtained are of good quality. Figures are also helpful. Therefore, the paper could be accepted for publication after revision. The following comments and suggestions are raised for the authors’ reference:

1.    The research significance and subsequent impact of this study on the state of the practice is suggested to be highlighted.

2.    The authors are invited to emphasize the innovation introduced by their numerical study over that present in the literature.

3.    The mesh selection is not justified in the description of the FEM. The influence of the mesh on the accuracy of the numerical results is not demonstrated adequately. Was there any parametric analysis conducted?

4.    ANSYS itself or any other finite element software does not simulate any material or structural element; it just provides the means to help the user proceed with any simulation. The user provides the input parameters, the appropriate geometry, load, type of finite elements, and boundary conditions. Thus, it is essential when a study is performed using finite element software to provide all of the above. Then, experimental results are used to validate the material constitutive laws and all simulation input parameters. Some comments concerning this issue are suggested to be added.

5.    Conclusions are long and could be shortened.

The English need some polishing since the phraseology seems cumbersome in several places. The language and the overall style of the presentation needs revision. For example:

- “we simulate”, “we went…”, “we can…”, etc. should be avoided.

- “MPa” should be used.

- Several references and citations using only one statement (such as [12-19]) should be avoided.

Author Response

I have sent responses to your review

Round 2

Reviewer 1 Report

  • The author has made the required changes and agreed to publish them.

Reviewer 2 Report

The questions and requirements posed by the previous review round have been addressed acceptably. As mentioned before, the paper is interesting, falls within the scope of the Journal, the manuscript is well-structured, figures are helpful, and the findings obtained are of good quality. Further, the revised manuscript with the Manuscript ID: “jcs-2488585-v2” and the title: “Topology optimization of a femoral stem in Titanium and Carbon to reduce Stress Shielding with the FEM method” has been improved, hence, the paper is suggested to be accepted for publication in the journal without further re-review.